# Current Strategies for Studying the Natural and Synthetic Bioactive Compounds in Food by Chromatographic Separation Techniques

Wioletta Parys *, Małgorzata Dołowy and Alina Pyka-Pająk *

Department of Analytical Chemistry, Faculty of Pharmaceutical Sciences in Sosnowiec,
Medical University of Silesia in Katowice, Jagiellońska 4, 41-200 Sosnowiec, Poland; mdolowy@sum.edu.pl
* Correspondence: wparys@sum.edu.pl (W.P.); apyka@sum.edu.pl (A.P.-P.); Tel.: +48-32-364-15-34 (W.P.);
+48-32-364-15-30 (A.P.-P.)

**Abstract:** The present study summarizes the new strategies including advanced equipment and validation parameters of liquid and gas chromatography methods i.e., thin-layer chromatography (TLC), column liquid chromatography (CLC), and gas chromatography (GC) suitable for the identification and quantitative determination of different natural and synthetic bioactive compounds present in food and food products, which play an important role in human health, within the period of 2019–2021 (January). Full characteristic of some of these procedures with their validation parameters is discussed in this work. The present review confirms the vital role of HPLC methodology in combination with different detection modes i.e., HPLC-UV, HPLC-DAD, HPLC-MS, and HPLC-MS/MS for the determination of natural and synthetic bioactive molecules for different purposes i.e., to characterize the chemical composition of food as well as in the multi-residue analysis of pesticides, NSAIDs, antibiotics, steroids, and others in food and food products.

**Keywords:** bioactive compounds; food; separation techniques; liquid chromatography; gas chromatography

## 1. Introduction

Bioactive compounds are natural or synthetic (partially or totally) compounds that show biological activity i.e., have the ability to interact with living tissues and indicate an effect on human body including the promotion of good health, thus they are important as new ingredients of the current functional food (e.g., antioxidants) [1]. Food samples are very complex mixtures consisting not only of naturally occurring bioactive compounds with beneficial role on human health like for example vitamins, minerals, antioxidants but other substances coming from agrochemical treatments i.e., pesticides as well as promotors animals growth or veterinary drugs. Therefore monitoring the level of different veterinary drugs or organic pesticides coming from agrochemical treatments in food and food products could ensure the safety of potential consumers. Natural and synthetic bioactive compounds occur in foods in small quantities and represent a wide group of chemical compounds. Because of the complexity of food matrices, the separation and next accurate determination of their bioactive constituents with different chemical structure requires an universal analytical methodology like liquid and gas chromatography or combination of both chromatographic techniques.

For this fact, this article reviews new strategies including advanced equipment and validation parameters of liquid and gas chromatography methods dedicated for the identification and quantitative analysis of natural and synthetic bioactive compounds occurring in food and food products within the period of 2019–2021 (January). Special attention is given to optimization including the validation process of chromatographic analysis performed by using thin-layer chromatography (TLC), high-performance liquid chromatography

(HPLC), and gas chromatography (GC) coupled with different detection modes ((TLC-UV/Vis, TLC-densitometry, HPTLC-MS, HPLC-UV/Vis, HPLC-DAD(PDA), HPLC-MS, HPLC-MS/MS, HPLC-TQ-ESI-MS/MS, GC-MS, GC-MS/MS, GC-CPI-MS/MS)) as well as the combined chromatographic techniques e.g., HPLC/GC that may be valuable for the separation, screening, quantitative determination or evaluation of certain physicochemical and pharmacological properties of many including the newly developed natural and synthetic bioactive compounds in food and food products.

## 2. Thin Layer Chromatography

Liquid chromatography, including thin-layer chromatography, along with other chromatographic techniques, is one of the most popular methods used in the current analysis of bioorganic and bioinorganic compounds in different including food samples [2–15].

*TLC Analysis of Selected Bioactive Compounds in Food Samples*

The recently published papers indicate that thin-layer chromatography was successfully used for the quantification of selected antibiotics, alkaloids, aromatic amines, and gallic acid in food [2–5]. Both i.e., contact and immersion TLC-bioautography with the use of silica gel $F_{254}$ plates, 7.5% of $KH_2PO_4$, and *Escherichia coli* ATCC 8739 as a test bacterium were employed for the sensitive determination of streptomycin in the presence of kanamycin sulfate in frozen shrimp, thus to control the antibiotic abuse in frozen food [2]. The work of Foudah et al. [3] shows a rapid and sensitive HPTLC method with densitometry for the quantification of trigonelline content as important bioactive constituent of Arabic coffees at the level of ng/spot [3]. Another study [4] indicates the use of HPTLC-DPPH (high-performance thin-layer chromatography coupled with the use of 2,2-diphenyl-1-picrylhydrazyl) method for rapid and simple screening of antioxidant constituents i.e., gallic acid in honey, in natural food products. Similarly, the study of Piszcz and coworkers [6] demonstrated the ability of TLC method to separate two different forms of DPPH (i.e., DPPH and DPPH-H) and also for the measurement of total antioxidant potential in the meat samples. Another study describes a novel and fully validated HPTLC-MS method for the rapid identification and determination of toxic aryl azo amines in food matrices. The achieved level of detection and quantification of these compounds was in ppm [5].

Another authors; Turkmen and Kurada [7] confirmed the utility of HPTLC on silica gel $60F_{254}$ plates with densitometric measurements to asses next toxic compound, namely patulin as contamination of fruit-based baby foods in Turkey.

In the vast majority of analyzes, fatty acids are investigated using the GC technique as fatty acid methyl esters. However, Dąbrowska et al. [8] developed a TLC method in combination with densitometry for the determination of omega-3 fatty acids: linolenic (ALA), docosahexaenoic (DHA), and eicosapentaenoic acids (EPA) in 15 dietary supplements and 5 cooking products.

Some studies indicate the important role of TLC and HPTLC methods as comprehensive techniques for the detection and identification of pesticides and the toxicity caused by these compounds [9–15]. Several new chromogenic reagents have been reported in the literature such as diphenylamine reagent for detection of organochloro insecticide endosulfan [9], stannous chloride and hydrochloric acid (reducing reagent) followed by a sodium nitrite in hydrochloric acid (coupling reagent) and β-napthol in sodium hydroxide for the detection of herbicide oxyfluorten [10], chloranil reagent with nitric acid for detection of organophosporus insecticide monocrotophos [11], 4-amminoantipyrene reagent with potassium ferricyanide for detection and identification of 2,4-dichlorophenol, an intermediate of 2,4-D (2,4-dichlorophenoxyacetic acid) herbicide [12], cupric acetate reagent for detection of organophosphate insecticide profenofos [13], and cobalt thiocyanate reagent for detection of organophosporus herbicide glyphosate [14]. Hussain et al. [15] developed an HPTLC method for the determination of residues of various pesticides in brinjal samples

from a market of Pakistan. The authors showed that HPTLC can be an alternative method to HPLC for the detection of pesticide residues.

The scientific literature cited and discussed above indicates that TLC/HPTLC can be successfully used to detect and quantify a wide variety of synthetic and natural classes of bioactive compounds occurring in food and food products. There are many reports in the scientific literature combining TLC with a densitometry. However, there is an increase of works linking TLC with MS. Therefore, it seems that in the next few years there should be more scientific papers using TLC/MS.

## 3. Column Liquid Chromatography

Extensive review of literature published in the two last years indicates that high-performance liquid chromatography (HPLC) with different detection systems such as ultraviolet detector (HPLC-UV), photodiode array detector (HPLC-PDA), or coupled to mass spectrometry or tandem mass spectrometry called as HPLC-MS and HPLC-MS/MS respectively is a powerful analytical tool with many applications including food analysis [16–66].

*Column Liqiud Chromatography in Analysis of Selected Bioactive Compounds in Food Samples*

Due the widespread use of agricultural chemicals in food production, people are exposed to low levels of pesticide residues through their diets. Because the organic pesticides usually exist in very small amounts in food samples and have different chemical structure containing, for example, triazine, imidazolinone, phenyluracyl, or macrocyclic lactone structure, thus there is a need to develop efficient and sensitive CLC systems for the simultaneous determination of compounds that are dangerous to human health, present in food and food products which belong to one of the presented groups as well as to various groups (i.e., multiclass pesticides) [31,32]. Table 1 shows the utility of selected CLC procedures with validation parameters that have been applied in analysis of food samples [16–48]. The current literature review indicates that validated high performance liquid chromatography is a powerful analytical technique used to determine many single or multi-class pesticides present in different food matrices. Most developed methods were validated according to the European SANTE guidelines (SANTE/11945/2015, SANTE/11813/2017, SANTE/12682/2019) in terms of linearity, LOD, LOQ, accuracy, recovery, and precision, as shown in Table 1. As it can be observed, liquid chromatography is particularly appropriate for the analysis of polar, non-volatile, and/or thermally labile pesticides. Because of its high selectivity and sensitivity, HPLC and UHPLC in combination with MS/MS have mostly been used in this field especially to determine the insecticides and herbicides belonging to organophosphorus compounds, imidazolinone and pyridine carboxylic acid derivatives, and in study of samples containing multiclass pesticides [16–24,27,30–34]. However, in a few cases, i.e., triazine and phenylurea herbicides, the HPLC coupled with spectrophotometric detection HPLC-UV or DAD has also been applied [25,26,28]. Various kinds of stationary phases (columns) have been used in the HPLC determination of pesticides, mainly C18 [16–18,20,25–31], and also chiral [19,33], BEH HILIC [21], Hypercarb [22], Obelisc N HILIC [24,34], Acquity UPLC HSS T3 [23,32]. In general, water or water with formic acid or acetic acid or ammonium formate, acetonitrile, and methanol have been applied as mobile phases with gradient or isocratic elution, respectively.

**Table 1.** Column liquid chromatography in food analysis.

| Matrix/Compound | Chromatographic Conditions | Other Parameters | Refs. |
|---|---|---|---|
| **Variety Classes of Pesticides** | | | |
| **Insecticides** | | | |
| *Containing macrocyclic lactone structure* | | | |
| Porcine muscle, egg, milk, eel, flatfish, shrimp Spinosyn A (SPA), Spinosyn D (SPD), Temephos (TP), Piperonyl butoxide (PB) | LC-TQ-ESI-MS/MS Multiple reaction monitoring (MRM) mode Phenomenex Kinetex EVO C18 (150 × 2.1 mm, 2.6 μm) Eluent A: 0.1% formic acid in 10 mM ammonium formate in distilled water; Eluent B: methanol A:B (10:90, *v/v*) Flow rate: 0.2 mL/min | Linearity (μg/kg): 3.5 ÷ 35 (for SPA), 1.5 ÷ 15 (for SPD) 5 ÷ 50 (for TP, PB) LOD (μg/kg): 0.5 ÷ 0.8 (for SPA), 0.1 (for SPD) 1.1 ÷ 1.6 (for TP), 0.3 ÷ 0.7 (for PB) Recovery: 70 ÷ 105% | [16] |
| *Organothiophosphate derivatives* | | | |
| Tomato, cabbage, barley, Xijiang river water, tap water Quinalphos(QP), Triazophos (TZ), Parathion (PTN), Fenthion (FT), Chlorpyrifos-methyl (CHM) | HPLC-UV λ = 254 nm Agilent TC-C18 (150 × 4.6 mm, 5 μm) Pure methanol Flow rate: 0.5 mL/min | Linearity: 0.02 ÷ 2.00 μg/mL LOD (μg/L): 3.0 (for QP), 5.0 (for TZ, PTN) 6.0 (for FT), 10.0 (for CHM) Recovery: 80 ÷ 98% | [17] |
| **Herbicides** | | | |
| *Phenoxyacetic acid derivatives* | | | |
| Corn, wheat, rice Phenoxy acid herbicides (6) | HPLC-TQ-ESI-MS/MS Multiple reaction monitoring (MRM) mode RP C18 (150 × 2.1 mm, 3.5 μm) Eluent A: water; Eluent B: acetonitrile Gradient elution Flow rate: 0.3 mL/min | Linearity: 0.200 ÷ 40.0 μg/kg LOD: 0.0500 ÷ 0.300 μg/kg Accuracy: 95.6 ÷ 107% Intraday precision: 0.895 ÷ 5.40% Interday precision: 1.13 ÷ 6.61% Recovery: 73.8 ÷ 115% | [18] |
| *Imidazolinone derivatives* | | | |
| Soybean, peanut, wheat, maize, rice *S*-imazethapyr (SIT) *R*-imazethapyr (RIT) *S*-imazamox (SIZ) *R*-imazamox (RIZ) *S*-imazapic (SIP) *R*-imazapic (RIP) | UPLC-TQ-ESI-MS/MS Multiple reaction monitoring (MRM) mode Chiralcel OJ-3R (150 × 4.6 mm, 3 μm) Eluent A: 0.1% formic acid aqueous solution; Eluent B: acetonitrile Gradient elution Flow rate: 0.4 mL/min | LOD (μg/kg): 0.35 ÷ 0.48 (for SIP), 0.36 ÷ 0.72 (for RIP) 0.40 ÷ 0.88 (for SIT), 0.34 ÷ 0.75 (for RIT) 1.0 ÷ 1.5 (for SIZ), 0.98 ÷ 1.4 (for RIZ) Recovery: 64.2÷106.4% | [19] |
| *Pyridine carboxylic acid derivatives* | | | |
| Milk aminopyralid, picloram, fluroxypyr, clopyralid | LC-TQ-ESI-MS/MS Multiple reaction monitoring (MRM) mode Waters Xselect HSS T3 (C18) (2.1 × 150 mm, 5 μm) Eluent A: ultrapure water; Eluent B: methanol Gradient elution Flow rate: 300 μL/min | Linearity: 1 ÷ 50 μg/L LOD: 0.124 μg/L Recovery: 75.3 ÷ 89.8% | [20] |

**Table 1.** *Cont.*

| Matrix/Compound | Chromatographic Conditions | Other Parameters | Refs. |
|---|---|---|---|
| *Quaternary ammonium salt derivatives* | | | |
| Barley, wheat Paraquat (PQ), Diquat (DQ), Chlormequat CHQ), Mepiquat (MQ) | UHPLC-TQ-ESI-MS/MS Selected reaction monitoring (SRM) mode Acquity UPLC$^{TM}$ BEH HILIC (100 × 2.1 mm, 1.7 μm) Eluent A: aqueous solution of ammonium formate 60 mmol/L at pH 3.7; Eluent B: acetonitrile A:B (40:60, *v/v*) Flow rate: 0.250 mL/min | Linearity (μg/kg): 80 ÷ 1000 (for CHQ), 40 ÷ 1000 (for MQ) 20 ÷ 1000 (for PQ, DQ) LOD (μg/kg): 24 (for CHQ), 12 (for MQ), 6 (for PQ, DQ) Recovery: 93 ÷ 106% | [21] |
| *Organophosphorus compounds, chlorates* | | | |
| Fruits, vegetables, infant foods Glyphosate (GLY), Aminomethyl phosphonic acid (AMPA), Phosphonic acid (PHA), Fosetyl-Al (FAL), Chlorate (CHL), Perchlorate (PCH) | UHPLC-Q Orbitrap-ESI-MS/MS Thermo Scientific Hypercarb (3 × 100 mm, 5 μm) Eluent A: 0.4% formic acid in methanol; Eluent B: 0.4% formic acid in purified water A:B (95:5, *v/v*) Flow rate: 0.3 mL/min | Linearity: 0.001 ÷ 0.1 mg/L LOQ (mg/kg): 0.0004 (for PCH) 0.001 (for FAL) 0.002 (for CHL) 0.003 (for GLY, AMPA) 0.004 (for PHA) Recovery: 72 ÷ 116% | [22] |
| Corn Glyphosate, Glufosinate | UHPLC-ESI-QTRAP-MS Multiple reaction monitoring (MRM) mode Acquity UPLC HSS T3 (2 × 100 mm, 1.8 μm) Eluent A: 0.05% ammonia water; Eluent B: acetonitrile A:B (90:10, *v/v*) Flow rate: 0.2 mL/min | Linearity: 10.0 ÷ 500 ng/mL LOD: 0.0015 mg/kg Recovery: 90.3 ÷ 95.4% Intraday precision: 1.24 ÷ 3.35% Interday precision: 3.56 ÷ 6.06% | [23] |
| Vegetable milk, beer, wine Highly polar pesticides (14) including: glyphosate, glufosinate, ethephon, fosetyl and metabolites | LC-ESI-QTRAP-MS Multiple reaction monitoring (MRM) mode Obelisc N HILIC (150 × 2.1 mm, 5 μm) Eluent A: water with 1% formic acid; Eluent B: acetonitrile Gradient elution Flow rate: 0.5 mL/min | Linearity: 0.2 ÷ 50 ng/mL ILOD (instrumental LOD): 0.2 ng/mL Recovery: 70 ÷ 120% | [24] |
| *Triazine compounds/chlorinated anilide derivatives* | | | |
| White gourd, tomato, soybean milk Metribuzin, Simetryn, Propazine, Prometryne | HPLC-DAD λ = 222 nm Centurysil C18 (200 × 4.6 mm, 5 μm) Eluent A: acetonitrile; Eluent B: water A:B (55:45, *v/v*) Flow rate: 1.0 mL/min | Linearity: 0.3 ÷ 100.0 ng/g for white gourd and tomato Linearity: 0.5 ÷ 100 ng/mL for soybean milk LOD: 0.10 ÷ 0.20 ng/g for white gourd and tomato LOD: 0.15 ÷ 0.30 ng/mL for soybean milk | [25] |
| Beans Atrazine (AZ), Oxadiazon (OZ), Metazachlor (MZ), Propanil (P) | HPLC-DAD λ = 230 nm Aqilent Eclipse XDB-C18 (150×4.6 mm, 3.5 μm) Eluent A: water; Eluent B: acetonitrile Gradient elution Flow rate: 0.50 mL/min | Linearity: 0.1 ÷ 10 μg/mL LOD (μg/kg): 10.3 (for AZ) 2.4 (for OZ) 2.9 (for MZ) 3.8 (for P) Recovery: 90.7 ÷ 116.5% | [26] |

**Table 1.** *Cont.*

| Matrix/Compound | Chromatographic Conditions | Other Parameters | Refs. |
|---|---|---|---|
| *Acidic herbicides* | | | |
| Cucumber, orange Acidic herbicides (27) Phytohormones (8) | UHPLC-TQ-ESI-MS/MS Multiple reaction monitoring (MRM) mode Acquity UPLC BEH C-18 (100 × 2.1 mm, 1.7 μm) Eluent A: 1% acetic acid and 5% methanol in water; Eluent B: 1% acetic acid in methanol Gradient elution Flow rate: 0.35 mL/min | For all compounds: Linearity: 10 ÷ 150 μg/kg LOQ: 10 μg/kg Recovery: 86 ÷ 120% Intraday precision: 1 ÷ 20% Interday precision: 4 ÷ 20% | [27] |
| *Phenylurea derivatives* | | | |
| Soybean milk, tomato Metoxuron, Monuron, Chlortoluron, Monolinuron, Buturon | HPLC-DAD $\lambda$ = 254 nmCenturysil C18 (250 × 4.6 mm, 5 μm) Eluent A: water; Eluent B: acetonitrile A:B (52:48, *v/v*) Flow rate: 1.0 mL/min | Linearity: 0.30 ÷ 150.0 ng/mL for soybean milk Linearity: 0.20 ÷ 150.0 ng/g for tomato LOD: 0.10 ÷ 0.20 ng/mL for soybean milk LOD: 0.06÷0.15 ng/g for tomato Recovery: 86.0 ÷ 115.2% | [28] |
| *Phenyluracil derivatives* | | | |
| Orange, apple, grape, mango, banana, pear, peachTiafenacil and its six metabolites | UHPLC-TQ-ESI-MS/MS Multiple reaction monitoring (MRM) mode Waters CORTECS C18 (150 × 2.1 mm, 2.7 μm) Eluent A: water containing 0.1% formic acid; Eluent B: acetonitrile Gradient elution Flow rate: 0.4 mL/min | Linearity: 5 ÷ 1000 μg/kg LOQ: 10 μg/kg Intraday precision (RSD): 1.0 ÷ 13.0% Interday precision (RSD): 1.1 ÷ 14.6% Recovery: 73 ÷ 105% | [29] |
| **Multiclass pesticides** | | | |
| Rice (*Oryza sativa* L.) Multiclass pesticides (155) | UHPLC-ESI-QTRAP-MS Multiple reaction monitoring (MRM) mode Fusion-RP 80A (50 × 2 mm, 4 μm) Eluent A: 0.1% formic acid aqueous solution; Eluent B: 0.1% formic acid in methanol Gradient elution Flow rate: 0.25 mL/min | Linearity: 5÷50 μg/L and 5 ÷ 60 μg/L LOQ: 5 μg/kg Recovery: 77.1 ÷ 111.5% | [30] |
| Pecan nuts Multiclass pesticides (47) | LC-TQ-ESI-MS/MS Selected reaction monitoring (SRM) mode Pursuit XRs Ultra C18 (100 × 2.0 mm, 1.7 μm) Eluent A: aqueous 5 mmol/L ammonium formate; Eluent B: methanol Gradient elution Flow rate: 0.150 mL/min | Linearity: 2.5 ÷ 125 μg/L LOD: 2 ÷ 3 μg/kg Recovery: 70 ÷ 120% | [31] |

**Table 1.** *Cont.*

| Matrix/Compound | Chromatographic Conditions | Other Parameters | Refs. |
|---|---|---|---|
| Sugarcane spirits (Brazilian cachaças) Multiclass pesticides (10) | UPLC-TQ-ESI-MS/MS Multiple reaction monitoring (MRM) mode Acquity UPLC HSS T3 (150 × 0.3 mm, 1.8 μm) Eluent A: water; Eluent B: acetonitrile Gradient elution Flow rate: 8 μL/min | Linearity: not given LOD: 5 μg/L Accuracy: 80 ÷ 123% Intraday precision (RSD): 0.31 ÷ 44.17% Interday precision (RSD): 0.23 ÷ 22.78% | [32] |
| **Other pesticides** | | | |
| Cucumber, tomato, cabbage, grape, mulberry, apple, pear Chiral pesticides (22) | LC-TQ-ESI-MS/MS Multiple reaction monitoring (MRM) mode Chiralpak IG (250 × 4.6 mm, 5 μm) with a Chiralpack IG guard column (10 × 4 mm, 5 μm) Eluent A: acetonitrile; Eluent B: ultrapure water containing 5 mmol/L ammonium acetate and 0.1% formic acid A:B (65:35, *v/v*) Flow rate: 0.6 mL/min | Linearity: 1 ÷ 200 ng/g ILOQ (instrumental LOQ): 0.33 ÷ 1.50 ng/g MLOQ (method LOQ): 0.15 ÷ 1.00 ng/gRecovery: 84.0 ÷ 112.3% Intraday precision (RSD): 2.3 ÷ 10.9% Interday precision (RSD): 3.0 ÷ 11.2 % | [33] |
| Grapes, lettuce, orange, oat, soya bean Highly polar pesticides (14) | LC-ESI-QTRAP-MS Multiple reaction monitoring (MRM) mode HILIC-column, Obelisc N (2.1 × 150 mm, 5 μm) Eluent A: water with 1% formic acid; Eluent B: acetonitrile Gradient elution Flow rate: 500 μL/min | Linearity: 0.1 ÷ 100 ng/mL LOQ: 0.02 ÷ 0.5 mg/kg Recovery: 70 ÷ 120% | [34] |
| **Non-steroidal anti-inflammatory compounds** **(derivatives of phenylpropionic acid, phenylacetic acid and acetylsalicylic acid) and chloramphenicol** | | | |
| Bovine milk, ovine milk NSAIDs: Carprofen (CPF),Tolfenamic acid (TFA), 5-hydroxy flunixin (HFX), Diclofenac (D), 4-methylaminoantipyrin (MAAP), Meloxicam (MX), Ibuprofen (I), Phenylbutazone (PBZ); antibiotic: Chloramphenicol (CHP) | LC-ESI-QTRAP-MS Scheduled multiple reaction monitoring (sMRM) mode Kinetex XB-C18 (100 × 2.1 mm, 2.6 μm) Eluent A: water containing 0.1% formic acid; Eluent B: methanol Gradient elution Flow rate: 0.4 mL/min | For all compounds:LOQ (μg/kg): 0.05 (for D) 0.15 (for CHP) 2.5 (for PBZ) 5 (for I) 7.5 (for MX) 20 (for HFX) 25 (for TFA, MAAP) 250 (for CPF) Accuracy: 87 ÷ 108% Interday precision (CV): 3 ÷ 16% | [35] |
| Bovine milk Diclofenac (D), Flurbiprofen (FB), Ketoprofen (KP), Mefenamic acid (MA) | HPLC-TQ-ESI-MS/MS Luna C18 (250 × 2.0 mm, 5 μm) Eluent A: methanol; Eluent B: 0.05% aqueous solution of formic acid A:B (3:1, *v/v*) Flow rate: 0.4 mL/min | Linearity (μg/kg): 0.03 ÷ 200 (for D, KP) 0.03 ÷ 300 (for FB) 0.1 ÷ 250 (for MA) LOD (μg/kg): 0.01 (for D, KP, FB) 0.03 (for MA) Recovery: 96 ÷ 107% | [36] |

<div align="center">Table 1. <i>Cont.</i></div>

| Matrix/Compound | Chromatographic Conditions | Other Parameters | Refs. |
|---|---|---|---|
| Meat of swine, chicken and bovine Multiclass NSAIDs (47) | LC-TQ-ESI-MS/MS Hypersil Gold C18 (150 × 2.1 mm, 5 μm) Eluent A: 0.1% formic acid with 0.5 mmol/L ammonium acetate; Eluent B: acetonitrile | Linearity: 0.1 ÷ 50 ng/mL LOD: 0.1 ÷ 0.5 ng/g Intraday precision (RSD): 2.2 ÷ 5.6% Interday precision (RSD): 5.3 ÷ 12.6% Recovery: 72.4 ÷ 97.1% | [37] |
| Bovine milk Veterinary drugs: Acetanilide (AAN), Anthranilic acid (ANA), Antipyrine (AP), Cyproheptadine (CHD), Diphenhydramine (DH), DL-methylephedrine (ME), Phenacetin (PA) | LC-TQ-ESI-MS/MS Multiple reaction monitoring (MRM) mode Waters Xbridge C18 (150 × 2.1 mm, 3.5 μm) Eluent A: 0.1% formic acid in water; Eluent B: 0.1% formic acid in acetonitrile Gradient elution Flow rate: 0.2 mL/min | Linearity: 1 ÷ 40 μg/kg LOD (μg/g): 0.3 (for AP) 0.4 (for CHD, ME) 0.5 (for DH) 0.6 (for PA) 2.1 (for AAN, ANA) Recovery: 71.2 ÷ 103.8% Intraday precision (RSD): 0.7 ÷ 6.4% Interday precision (RSD): 0.1 ÷ 8.6% | [38] |
| Fish tissues Ibuprofen, Indoprofen, Pranoprofen, Flurbiprofen, Ketoprofen, Carprofen, Naproxen, Loxoprofen, Etodolac | UHPLC-TQ-ESI-MS/MS Multiple reaction monitoring (MRM) mode Chiralpak ID (250 × 4.6 mm, 5 μm) with guard column (10 × 4.6 mm, 5 μm) Eluent A: 40% acetonitrile Eluent B: water containing 20 mM $HCOONH_4$ Gradient elution Flow rate: 0.6 mL/min | Linearity: 2 ÷ 400 ng/g LOD: 1 ÷ 8 ng/g Recovery: 82.6 ÷ 106.7% Intraday precision (RSD) ≤ 8.2% Interday precision (RSD) ≤ 8.2% | [39] |
| Meat, egg Ibuprofen (I), Naproxen (N), Diclofenac (D), Carprofen (CPF), Ketoprofen (KP), Tolfenamic acid (TFA), Salicylic acid (SA) | UPLC-TQ-ESI-MS/MS Multiple reaction monitoring (MRM) mode Acquity UPLC BEH C18 (50 × 2.1 mm, 1.7 μm) Eluent A: methanol; Eluent B: water with 0.1% formic acid Gradient elution Flow rate: 0.25 mL/min | Linearity (μg/kg): 5 ÷ 1500 (for D), 10 ÷ 1500 (for N) 20 ÷ 1500 (for CPF, KP, SA) 30 ÷ 1500 (for TFA), 40÷1500 (for I) LOD (μg/kg): 9.1 ÷ 12.2 (for I), 2.1 ÷ 2.4 (for N) 1.2 ÷ 1.4 (for D), 5.7÷6.0 (for CPF, KP) 7.5 ÷ 10.7 (for TFA), 4.5 ÷ 5.6 (for SA) Intraday precision (RSD): 4.06 ÷ 16.01% Interday precision (RSD): 2.74 ÷ 14.25% Recovery: 85.18 ÷ 109.8% | [40] |
| **Antibiotics (fluoroquinolones)** | | | |
| Chicken meat Beef meat Feroxacin (FRX), Ofloxacin (OF) | HPLC-FLD λ = 278 nm and 466 nm Luna C18 (250 × 4.6 mm, 5 μm) Eluent A: Methanol Eluent B: 0.05 mol/L phosphate buffer (pH = 6.4) Gradient elution Flow rate 0.7 mL/min at 30 °C | For FRX: Linearity: 40 ÷ 4000 μg/kg LOD: 15 μg/kg, LOQ: 40 μg/kg Recovery: 98 ÷ 108% For OF: Linearity: 30 ÷ 3000 μg/kg LOD: 10 μg/kg, LOQ: 30 μg/kg Recovery: 100 ÷ 107% | [41] |

**Table 1.** *Cont.*

| Matrix/Compound | Chromatographic Conditions | Other Parameters | Refs. |
|---|---|---|---|
| **Steroid compounds** | | | |
| Meat samples of different categories (chicken, beef, sheep, camels) Some estrogens: estrone (E1), 17β-estradiol (E2), estriol (E3), natural estrogens and 17-α ethinylestradiol (E4) an exoestrogen | HPLC-DAD, λ = 220 nm Symmetry C18 (4.6 × 150 mm, 3.5 μm) Eluent A: acetonitrile Eluent B: water A:B: (50:50, *v/v*) Flow rate: 1 mL/min | LOD (μg/g): 0.126 (for E1, E2) 0.094 (for E3, E4) LOQ (μg/g): 0.350 (for E1, E2) 0.188 (for E3, E4) | [42] |
| Samples of chicken egg white Corticosterone | HPLC-MS/MS Agilent Zorbax Eclipse Plus C18 (2.1 × 100 mm, 1.8 μm) Eluent A: 0.1% formic acid in water Eluent B: acetonitrile-0.1% formic acid Gradient elution Flow rate: 0.4 mL/min | LOQ: 0.02 ng/mL Recovery: 48.1% | [43] |
| Samples of Antarctic krill (*Euphausia superba Dana*) 17 Endogenous and exogenous steroid hormones | UHPLC-MS Acchrom Unitary C18 (2.1 × 150 mm, 5 μm) Eluent A: water containing 0.1% formic acid Eluent B: methanol Gradient elution Flow rate: 0.2 mL/min | LOD: 2 ÷ 30 ng/kg, LOQ: 10 ÷ 100 ng/kg Recovery: 75.4 ÷ 110.6% | [44] |
| **Antioxidants (polyphenols and related compounds)** | | | |
| Samples of various food consumed in Malaysia, such as chewing gum, noodle, snacks, nut, chocolate, fruit juices, coffee, oat, biscuit Synthetic phenolic antioxidants (SPAs): propyl gallate, tert-butylhydroquinone, butylated hydroxyanisole, and butylated hydroxytoluene | HPLC-DAD, λ = 280 nm Agilent ZORBAX Eclipse XDB 5 μm C18 (150 mm × 4.6 mm, 5 μm) Eluent A: ultrapure water Eluent B: acetonitrile Gradient elution Flow rate: 2.0 mL/min | Linearity: 1 ÷ 300 mg/L LOD: 0.02 ÷ 0.67 mg/L, LOQ: 0.06 ÷ 2.03 mg/L Precision: 0.15 ÷ 0.84% Recovery: 80.4 ÷ 119.0% | [45] |
| Milk samples from dairy cows Quercetin | UHPLC-MS/MS ZORBAX SB-C18 (50 × 2.1 mm × 1.8 μm) Eluent A: methanol Eluent B: 0.5% formic acid Gradient elution Flow rate: 0.5 mL/min | LOQ: 1.0 μg/kg Intraday precision: <10% Interday precision: <15% Repeatability: 3 ÷ 7.2% Reproducibility: 6.1 ÷ 12% Recovery: 98% | [46] |
| Samples of green coffee produced company from Skopje, Macedonia Chlorogenic acid | RP-HPLC-DAD λ = 325 nm Poroshell 120 EC-C18 (50 × 3 mm, 2.7 μm) Eluent A: acetonitrile Eluent B: water with 1% phosphoric acid A:B (10:90, *v/v*) Flow rate: 1 mL/min | Linearity: 12.33 ÷ 143.50 μg/mL LOD: 0.29 pg LOQ: 0.96 pg Intraday precision (RSD peak area): 0.19% (RSD height): 1.32% Recovery: 97.87 ÷ 106.67% | [47] |

| Matrix/Compound | Chromatographic Conditions | Other Parameters | Refs. |
|---|---|---|---|
| Samples of commercially available red wines from Serbia 16 selected phenolic compounds: gallic acid (GA), p-hydroxybenzoic acid (HBA), catechin (CAT), syringic acid (SGA), trans-cinnamic acid (TCA), hesperetin (HP), naringenin (NG), vanillic acid (VA), benzoic acid (BZA), coumaric acid (CMA), resveratrol (RV), chlorogenic acid (CGA), caffeic acid (CFA), rutin (RN), quercetin (Q), kaempferol (KF) | HPLC-DAD $\lambda = 280$ nm (GA, HBA, CAT, SGA, TCA, HP, NG) $\lambda = 225$ nm (VA, BZA, CMA, RV) $\lambda = 360$ nm (KF) Poroshell 120 EC-C18 ($4.6 \times 100$ mm, 2.7 μm) Eluent A: distilled water with 0.1% glacial acetic acid Eluent B: acetonitrile with 0.1% glacial acetic acid Gradient elution Flow rate: 1.0 mL/min | Linearity (mg/L): 2.5 ÷ 25 (for CAT, VA) 1.0 ÷ 25 (for other compounds) LOD (mg/L): 0.03 (for RV) ÷ 0.62 (for CAT) LOQ (mg/L): 0.11 (for RV, TCA) ÷ 2.08 (for CAT) Recovery: 96.5 ÷ 100.9% | [48] |

It is commonly known that the HPLC-UV (DAD) technique has a lower sensitivity compared to the LC-MS/MS. However, owing to the new SPE (solid phase extraction) systems consisting novel polymers as adsorbents e.g., porous organic polymer Car-DMB, Py-DMB HCP (heterocyclic hypercrosslinked polymer), HPLC analysis further allows the quantification of some pesticides in food samples at concentrations of ng/g [25,28]. As is shown in Table 1, many HPLC-MS/MS techniques with triple quadrupole (TQ), electrospray ionization (ESI) in multiple reaction monitoring (MRM) mode or selected reaction monitoring (SRM) mode have been mainly used for the determination of different kind of pesticides [16,18–21,27,31,33]. In addition, the HPLC-MS/MS methods with electrospray ionization (ESI) and quadrupole trap (QTRAP) in multiple reaction monitoring (MRM) mode have been also employed in the analysis of various pesticides [23,24,30,34]. Whereas UHPLC-Q Orbitrap-ESI-MS/MS has been applied for the determination of highly polar pesticides and contaminants (glyphosate, aminomethyl phosphonic acid (AMPA), phosphonic acid, fosetyl-Al, chlorate, and perchlorate) in processed fruits, vegetables, and infant foods [22].

Studies [19,33] indicate that chiral LC-TQ-ESI-MS/MS and UPLC-TQ-ESI-MS/MS in MRM mode have been successfully applied to the simultaneous enantioselective determination of chiral pesticides in different vegetables and fruits. Martínez et al. [27] determined 27 acidic herbicides and 8 phytohormones in fruits and vegetables using UHPLC-TQ-ESI-MS/MS technique in the MRM mode.

Several papers created during the last two years [35–41] demonstrate the importance of different CLC procedures to determine selected veterinary drugs in animal food and food products belonging to various groups including non-steroidal anti-inflammatory agents (NSAIDS), some antibiotics, and others according to EU Commission Decision 2002/657/EC requirements [35] to guarantee food safety.

Whereas, LC-MS/MS methods with triple quadrupole (TQ), electrospray ionization (ESI) in multiple reaction monitoring (MRM) mode have been used for the determination of multiclass NSAIDs in meat of swine, chicken, eggs, and bovine [37,38,40]. Developed chiral UHPLC-TQ-ESI-MS/MS in MRM mode have been successfully applied to the simultaneous determination of four profens enantiomers including naproxen, carprofen, indoprofen, and flurbiprofen in fish tissues [39]. The obtained LODs and LOQs for each enantiomer ranged from 1 to 8 ng/g and 2 to 10 ng/g, respectively [39].

Kurjogi et al. [49] applied an HPLC-UV for the detection of antibiotics in milk samples originating from the dairy herds located in India. Similarly, Dinh et al. [50] elaborated QuEChERS-LC-MS/MS clean up method with UHPLC-MS/MS for the analysis of sulfonamides and potentiators, macrolides, lincosamides, quinolones and fluoro-

quinolones, nitrofurans, nitroimidazoles, chloramphenicol, triphenyl-methane dyes, tera-cyclines, and metabolites in cultured and wild seafood sold (in red-meat fish, white-meat fish, and shrimp).

Studies confirm the vital role of HPLC with diode array detection method and mass spectrometry for the analysis of some steroids in current residual food analysis of meat products and eggs coming from farmed animals, thus to control steroids in meat [42,43]. A reliable and sensitive UHPLC-MS method was also constructed by Han and Liu to detect 17 endogenous and exogenous steroid hormones including estrogens, androgens, glucocorticosteroids, and mineralocorticosteroids in Antarctic krill (*Euphausia superba Dana*) [44].

Another study shows the utility of HPLC with MS/MS based on the operation of a triple quadrupole (LC-ESI-MS/MS) for quality control of the species of meat or products by determining the presence of thermostable dipeptides (e.g., anserine, carnosine and balenin) [51].

Some studies demonstrate the important role of HPLC with UV, DAD, or FL detector as well as UHPLC-MS/MS in the study of patulin (mycotoxin) and related compounds in fruits e.g., mangoes, apples, grapes, oranges, and fruit products (juices and drinks) for children [52–58]. In this case C18 column and different usually binary mobile phases consisting, for example, of eluent A: 10 mM ammonium acetate in water and eluent B: 10 mM ammonium acetate in methanol [52] or acetonitrile-water [54] with gradient elution have been successfully applied. These methods allowed determining patulin at different levels given in µg/mL or µg/kg [52–58].

Several authors have also described the analytical methodologies based on HPLC to characterize the food composition i.e., to detect especially a new bioactive compounds with nutritional value and a proper biological activity, for example, antioxidant properties that are present in vegetables and fruits consumed in various countries. Developed methods are necessary to control the quality/authenticity of food and have been carried out by researchers during the last two years.

Numerous studies indicate that HPLC is the method of choice due to its precision and sensitivity for the determination and quantification of natural as well as synthetic antioxidants in various food/food products [45–47,59–64]. The main group of antioxidants investigated were phenolic compounds, especially phenolic acids, catechins, and flavonoids. Therefore the identification and assessment of antioxidant activity of different edible plant samples containing these bioactive compounds and their derivatives using high-performance liquid chromatography have been extensively investigated in the two last years. For example Yue et al. [45] developed and validated an HPLC-DAD method for the identification of selected synthetic phenolic antioxidants (SPAs) in chewing gum, noodle, snacks, nut, chocolate, fruit juices, coffee, oat, and biscuits. An interesting study performed by Cheung et al. [59] shows the utility of this technique for the determination of phenolic acids (16) and flavonoids (14) profiles in honey samples, thus for quality control of honey.

Gbylik-Sikorska et al. [46] developed for the first time an UHPLC-MS/MS method for the estimation of the pharmacokinetic parameters of quercetin in milk samples of dairy cows.

A few papers indicate the HPLC studies of different phenolic compounds in green coffee and the fruits of the three European plum cultivators [47,60].

Pepe et al. [61] undertook the study of the composition of polyphenols (26) and an-thocyanins (12) found in *Citrus sinensis* and *Vitis vinifera*. RP-UHPLC-PDA combined with LCMS-IT-TOF (ion trap-time of flight mass spectrometer) was used in analysis of polyphenols and anthocyanins. HPLC with UV-Vis detection was also used for the determination of anthocyanin in skins and seeds of five Greek red grape varieties [62].

Similar study by means of HPLC-MS/MS method was performed to estimate the contents of some antioxidant components in grapevine seeds *Vitis vinifera* L cultivated in Italy [63]. The results of chromatographic analysis confirmed the presence of nine major flavonoids (apigenin, astragalin, hyperoside, isorhamnetin, kaempferol, myricetin,

quercetin, quercitrin, and rutin) and two procyanidins (procyanidin A$_2$ and procyanidin B) in the studied extracts.

Carotenoids and polyphenols were evaluated and quantified by HPLC-DAD and UHPLC-Q-Orbitrap HRMS, respectively, in two-pigmented *Lactuca sativa L. var.* [64]. Separation and quantification of carotenoids were performed by HPLC-DAD on C18 column. Polyphenols analysis was performed by UHPLC-Q-Orbitrap HRMS on biphenyl column. LODs and LOQs of analyzed compounds were in the range of 0.03–0.05 and 0.10–0.16 ng/g, respectively.

Another author Cirilli et al. [65] investigated iberin (an isothiocynate with chemoprevention of different tumors) in natural products and in different food supplements. Analysis was performed by UHPLC-PDA-ESI/MS. Three degradation products of iberin were identified, namely: thiourea, methyl thiocarbamate, and ethyl thiocarbamate. Similar study refers to 6-methoxymellein as the main ingredient responsible for the bitterness of carrot (*Daucus carota* L.) [66].

Summarizing, it can be stated that the studies described above confirm that validated high-performance liquid chromatography methods coupled with DAD, UV-Vis, MS/MS, and HPLC-TQ-ESI-MS/MS are the powerful tools in analysis i.e., separation, identification, and quantification of different natural and synthetic bioactive compounds occurring in food and food products for different purposes, i.e., authenticity and safety of food and food products.

It was stated that examined by column liquid chromatography bioactive compounds in food samples belonged to different chemical classes e.g., steroids, phenolic compounds, variety antibiotics (fluoroquinolones, tetracyclines, β-lactams), organophosphorus, phenyluracyl or triazines pesticides, and others. Therefore, both the factors, chemical diversity and the complexity of investigated mixtures, i.e., the kind of studied matrix were the biggest challenges in the case of HPLC technique and were accurately described in this review paper. A broad variety of packing material of column including a new one such as molecularly imprinted magnetic polymers as well as modern extraction systems like solid-phase extraction and salting-out extraction combined with switchable-hydrophilicity solvent liquid–liquid microextraction to sample preparation allow separation and quantification of new bioactive compounds like synthetic antioxidants or trace levels of different chemical groups of pesticides simultaneously (i.e., multiclass pesticides) in food. The use of chiral stationary phases improves the separation and determination of the selected stereoisomers (S- and R-form) of some imidazolinonen herbicides in food samples (e.g., soybean, peanut, wheat, maize, rice) and some NSAIDs belonging to profens i.e., ibuprofen, indoprofren, pranoprofen, flurbiprofen, ketoprofen, caprofen, naproxen and loxoprofen in fish tissues simultaneously at the level of ng/g.

Properly validated for optimal conditions HPLC method by means of DAD (PDA) and UV-Vis detector with gradient elution program makes this technique enough sensitive for the quantitative determination of different bioactive compounds including the selected pesticides and drugs in food samples in µg/mL or ng/g, respectively.

## 4. Gas Chromatography

*GC in Analysis of Selected Bioactive Compounds in Food Samples*

Recent literature review shows that gas chromatography coupled to single or tandem mass spectrometric approaches (GC-MS, GC-MS/MS) served as an efficient tool for the determination of various organic compounds in food samples (Table 2). GC was used to quantify: 200 multiclass pesticides in fruits [67]; 14 lipophilic pesticides in raw propolis [68]; 5 organophosphorus pesticides (OPPs) in fruit juice and water [69], endocrine disrupting chemicals (EDCs) i.e., alkylphenols; 4 phenylphenols, bisphenol A; 7 parabens; 11 OPPs and triclosan in different cereal-based foodstuffs [70]; 4 isomers of hexachlorocyclohexane; 6 pyrethroid pesticides i.e., bifenthrin, fenpropathrin, cyhalothrin, cyfluthrin, cypermethrin, deltamethrin in milk [71]; 133 multiclass pesticides in pericarpium citri reticulatae (chenpi) [72]; 5 NSAIDs i.e., ibuprofen, paracetamol, diclofenac, naproxen, ketoprofen;

3 natural estrogens i.e., estrone, 17β-estradiol, estriol in Mussels *Mytilus edulis trossulus* [73], glyoxal and methylglyoxal in different alcoholic beverage and fermented foods [74], essential fatty acids in cereals and green vegetables [75], and fatty acids in grilled pork [76].

**Table 2.** GC in analysis of food samples.

| Matrix/Compound | Chromatographic Conditions | Other Parameters | Refs. |
|---|---|---|---|
| **Pesticides (organophosphorus and multiclass pesticides)** | | | |
| Banana, watermelon, pear, strawberry Multiclass pesticides (200) | GC-HRMS-Q-Orbitrap Agilent VF-5 MS (30 m × 0.25 mm, 0.25 μm) Carrier gas: helium Flow rate: 1.0 mL/min | Linearity: 1 ÷ 100 μg/kg LOQ: 5 μg/kg Recovery: 70 ÷ 120% Intraday and Interday precision (RSD): <20% | [67] |
| Raw propolis Lipophilic pesticides (14) | GC-EI-MS/MS Multiple reaction monitoring (MRM) mode Agilent HP-5 MS (30 m × 0.25 mm, 0.25 μm) Carrier gas: helium Flow rate: 1.0 mL/min | Linearity: 0.001 ÷ 0.200 μg/mL LOQ: 0.002 ÷ 0.020 μg/g Recovery: 61 ÷ 106.8% | [68] |
| Apple juice, grape juice, water Organophosphorus pesticides (OPPs): Phorate (PHT), Dimethoate (DMT), Diazinone (DZ), Disulfoton (DSF), Chlorpyrifos (CPF) | GC-EI-MS selected ion monitoring (SIM) mode Agilent HP-5 MS (30 m × 0.25 mm, 0.25 μm) Carrier gas: helium Flow rate: 1.0 mL/min | Linearity: 2.0 ÷ 500.0 μg/L LOD (μg/L): 0.9 (for PHT), 0.4 (for DMT), 0.6 (for DZ), 0.3 (for DSF), 1.0 (for CPF) Recovery: 83 ÷ 105% | [69] |
| Wheat flour, rice, spaghetti, cheese tortellini, macaroni, noodles, sesame regañas, wheat tortillas, corn flakes, crunchy fruit muesli, cookies, white bread, multiseed EDCs (Endocrine Disrupting Chemicals) (24): alkylphenols and phenylphenols (4), bisphenol A, parabens (7), pesticides (11), triclosan (personal care product) | GC-EI-MS selected ion monitoring (SIM) mode DB-5MS (30 m × 0.25 mm, 0.25 μm) Carrier gas: helium Flow rate: 1.0 mL/min | For all compounds: Linearity: 1.3 ÷ 2500 ng/kg LOD: 0.4 ÷ 23 ng/kg Intraday precision (RSD): 3.8 ÷ 6.2% Interday precision (RSD): 5.2 ÷ 7.2% Recovery: 82 ÷ 105% For pesticides: Linearity: 21 ÷ 2500 ng/kg LOD: 6.2 ÷ 23 ng/kg Intraday precision (RSD): 5.0 ÷ 6.2% Interday precision (RSD): 6.5 ÷ 7.2% Recovery: 83 ÷ 105% | [70] |
| Milk Isomers of hexachlorocyclohexane (α-HCH, β-HCH, γ-HCH, δ-HCH) and pyrethroid pesticides (bifenthrin, fenpropathrin, cyhalothrin, cyfluthrin, cypermethrin, deltamethrin) | GC-ECD ZB-5 (30 m × 0.25 mm, 0.25 μm) Carrier gas: nitrogen Flow rate: 0.72 mL/min | For all compounds: Linearity: 0.00143 ÷ 3.57 mg/L LOD: 0.07 ÷ 2 μg/kg LOQ: 0.2 ÷ 5 μg/kg Recovery: 70.1 ÷ 106.3% | [71] |
| Pericarpium citri reticulatae (chenpi) Multiclass pesticides (133) | GC-EI-MS/MS Multiple reaction monitoring (MRM) mode DB-5MS IU (30 m × 0.25 mm, 0.25 μm) Carrier gas: helium Flow rate: 1.5 mL/min | Linearity: 1 ÷ 200 ng/mL LOQ: 0.005 ÷ 0.01 mg/kg Recovery: 70 ÷ 112.2% | [72] |

**Table 2.** *Cont.*

| Matrix/Compound | Chromatographic Conditions | Other Parameters | Refs. |
| --- | --- | --- | --- |
| **Non-steroidal anti-inflammatory compounds (profens) and Steroids** | | | |
| Mussels *Mytilus edulis trossulus* NSAID (5): ibuprofen, paracetamol, diclofenac, naproxen, ketoprofen Natural estrogens (3): estrone, 17β-estradiol, estriol | GC-MS Selected ion monitoring (SIM) mode Zebron ZB-5MSi (30 m × 0.25 mm, 0.25 μm) Carrier gas: helium | For all compounds: LOD: 1 ÷ 7 ng/g Intermediate precision (RSD): 0.24 ÷ 9.82% Repeatability (RSD): 0.94 ÷ 7.82% Recovery: 80 ÷ 118% For NSAID: LOD: 1 ÷ 2 ng/g Intermediate precision (RSD): 0.69 ÷ 7.85% Repeatability (RSD): 0.94 ÷ 4.92% Recovery: 80 ÷ 115% | [73] |
| **Fatty acids** | | | |
| Cereals and green vegetables Essential fatty acids | ID-GC/MS HP-88 capillary column (60 m × 0.25 mm, 0.2 μm) Carrier gas: helium Flow rate: 1.0 mL/min | Repeatability (RSD): 0.23 ÷ 1.61% for the cereal samples 0.39 ÷ 1.89% for vegetable samples Repeatability for linoleic acid (RSD): 1.48 and 0.95% for rice and wheat flours Content of linoleic acid: 3614 mg/kg for rice flour 8402 mg/kg for wheat flour 6353 mg/kg for spinach powder 1353 mg/kg for Kimchi cabbage powder; Content of α-linolenic acid: 19786 mg/kg for spinach powder 9533 mg/kg for Kimchi cabbage powder | [75] |
| Grilled pork Fatty acids | GC-MS CP-Sil88 (100 m × 0.25 mm, 0.2 μm) Carrier gas: helium | LOQ: 0.1% of the total fatty acids Content of: Palmitic acid: 17.3 ÷ 55.4% Stearic acid: 8.8 ÷ 20.9% Oleic acid: 24.4 ÷ 48.8% Linoleic acid: 0.5 ÷ 3.6% Stearidonic acid: <0.1 ÷ 4.2% Docosahexaenoic acid: 0.5 ÷ 1.4% Gamma linolenic acid: <1% di-homo-γ- linolenic acid: <1% eicosapentaenoic acid: <1% | [76] |
| **Other compounds** | | | |
| Alcoholic beverage (wine, bear, makgeoli, soju, and fruit liquor) Fermented foods (soybean paste, red pepper paste, soy sauce) Glyoxal (GX), Methylglyoxal (MGX) | GC-MS HP-InnoWax capillary column (60 m × 0.25 mm, 0.25 μm) Carrier gas: helium Flow rate: 1.0 mL/min | For GLX: Working range 5 ÷ 4000 μg/kg Accuracy: 93.3 ÷ 104.5% Intraday precision: 4.3 ÷ 7.6% Interday precision: 3.0 ÷ 6.4% LOD: 1.1 μg/kg For MGX: Working range 5 ÷ 4000 μg/kg Accuracy: 92.9 ÷ 104.2% Intraday precision: 4.8 ÷ 7.9% Interday precision: 3.6 ÷ 7.5% LOD: 0.7 μg/kg | [74] |

Crude fat, total saturated acids, and total *trans* fatty acids in home meal replacements, and restaurant foods were analyzed using GC-FID (gas chromatography–flame ionization detector). Total crude fat contents were 0.61 ÷ 6.75 g/100 g, and 0.22 ÷ 5.69 g/100 g for home meal replacements and restaurant foods, respectively. Total saturated fatty acids contents were 0.08 ÷ 1.42 g/100 g, and 0.07 ÷ 1.44 g/100 g for home meal replacements and restaurant foods, respectively. Total *trans* fatty acids contents were 0.0 ÷ 0.11 g/100 g, and 0.0 ÷ 0.07 g/100 g for home meal replacements and restaurant foods, respectively [77]. Fatty acids in the form of methyl esters were also determined using the GC-FID technique in four bee products. The authors of the study compared the total fatty acid concentration (saturated, unsaturated, omega-3, omega-6, the ratio of saturated and unsaturated, omega-3/omega-6 fatty acids and trans fatty acids) [78]. Fruehwirth et al. [79] investigated the lipid oxidation in stored margarine using GC-FID method. Volatile components and fatty acids present in margarines were tested. Acetone and hexanal increased in all types of margarine during storage.

Study [80] shows the applicability of GC-MS analysis for identification of chemical components with different activity including antioxidant properties of varieties, not well described in literature, of edible plants and fruits cultivated in different countries. GC-MS was successfully applied for the separation and identification of chemical components with antioxidant activity such as different phenolic acids from citrus fruits cultivated in India i.e., grapefruits. The major components found were: limonene, methyl-cyclohexane, hexane-3-one, 3-hexanol, 2-hexanol, myrcene, sabinene, nonanal, neral, geranyl acetate, ostole. These compounds might contribute to the antioxidant activity of the juice and oil [80].

The reviewed papers confirm that gas chromatography has recently been used to study food and edible plants (the contents of pesticides, endocrine disrupting chemicals, NSAIDs, natural estrogens, glyoxal, methylglyoxal, fatty acids, compounds with antioxidant properties, such as e.g., flavonoids, phenolic compounds). The most commonly used gas chromatography was combined with a mass spectrometer or a dual mass spectrometer with electrospray ionization (GC-EI-MS, GC-EI-MS/MS). The presented papers show the utility of this technique for both, i.e., residue analysis of multiclass pesticides and NSAIDs simultaneously in food and food products as well as for the determination of new antibacterial and antitumor agents in edible plants.

## 5. Combined Techniques

In many cases, not one but two or more analytical techniques are required for determining the active substances present in food matrices. Nowadays, these combined techniques are powerful analytical tools with many applications. Several papers reported their utility in food analysis [81–86].

Carotenoids, phenolic compounds, and fatty acids were determined in tomato seed oil derived from cold break and hot break processing lines [81]. HPLC-DAD-ESI-MS on C18 column and two mobile phases in the gradient elution were used in the investigation of phenolic compounds. HPLC-DAD on C18 column and two mobile phases were used for the quantitative and qualitative analysis of carotenoids. Fatty acid profile was determined by GC-MS. Higher levels of carotenoids (lutein, lycopene, β-carotene) and phenolic compounds ((caffeic acid-glucoside isomer (CG), caffeic acid (CA), syringic acid (SyA), di-caffeoylquinic acid (di-CQA), and tri-Caffeoylquinic acid (tri-CQA)) were found in the cold pressed oil. The following fatty acids were the most abundant in the oil: linoleic acid, oleic acid, and palmitic acid [81].

Migas et al. [82] determined lutein and lutein mixed with zeaxanthin in eight dietary supplements. BMD-TLC (bivariant multiple development thin layer chromatography) was used for the analysis of lutein, β-carotene in samples. HPLC-DAD-ESI-MS was used for the isolation and identification of mixture of lutein and zeaxanthin. The proposed method was linear in the range 90 ÷ 500 ng/point. Limits of detection and quantification were 50 ng/point and 90 ng/point, respectively. Method was precise, accurate, and robust.

TLC was used for monitoring the formation of γ-aminobutyric acid (GABA) in traditional Indonesian foods fermented with thirty strains of lactic acid. For this purpose, silica gel 60F$_{254}$ plates and *n*-butanol-acetic acid-distilled water (5:2:2) mobile phase were used. On the other hand, for the quantitative determination of GABA, UPLC was used with the C18 column [83].

Aflatoxins are produced by fungi, including those on spoiled food. TLC on silica gel 60 plates using acetonitrile-methanol-trifluoroacetic acid (9:1:0.2, *v/v/v*) mobile phase and with the visualization using vanillin, p-anisaldehyde solutions, or iodine vapor was a simple, robust, and non-quantitative method for the detection of aflatoxins. HPLC-DAD ($\lambda$ = 200 ÷ 410 nm) with C18 column and two eluents in gradient elution were used for the quantitative determination of aflatoxins. TOF/Q-TOF MS/MS was used for the detection of aflatoxin metabolites, and the sixteen possible metabolites were identified [84].

A novel and highly sensitive metastable state nanoparticle-enhanced Raman spectroscopy combined with thin layer chromatography (TLC-MSNERS) has been successfully used for the determination of pesticides such as thiabendazole, phosmet, and triazophos on fruit skin. An amphiphilic polymer polyurethane-Ag nanoparticle (AgNPs) has been employed as the MSNERS substrate [85]. Another work developed and validated a modified QuEChERS method to determine multiclass pesticides (207) in honey samples using both LC-MS/MS (154 compounds) and GC-MS/MS (53 compounds) [86].

In summary, the necessity to analyze samples with a complicated composition requires the use of combined techniques. Sometimes the matrix is so complex (it contains chemical compounds belonging to different chemical classes) that there is a need to use at least two analytical techniques to determine the composition of the analyzed sample. The reliability requirements of the analytical results often preclude the possibility of identifying the analytes solely on the basis of the retention time. Only the combination of the ability to separate complex mixtures using chromatographic methods with structural information (HPTLC/MS, LC/MS, GC/MS) enables reliable identification of food constituents.

Owing to the use of combined techniques, it is possible to significantly speed up and reduce the cost of analyzing due to less requirements for the stage of sample preparation for analysis.

The advantages of the combined techniques in food analysis are: the ability to identify unknown food constituents, information about their molecular weight and/or structure, easy detection of the overlap between peaks, and faster end results. In contrast, the disadvantages of the combined techniques are high investment costs.

## 6. Conclusions

The reviewed papers confirm that of all chromatographic techniques, liquid chromatography (LC) is the most universal technique that enables successful analysis of complex matrices including food products. The current high-performance liquid chromatography systems are crucial to assess the quality of food. HPLC method in combination with various detection modes i.e., HPLC-UV, HPLC-DAD(PDA) and HPLC-MS or HPLC-MS/MS, respectively is selective, sensitive, accurate, and robust for the simultaneous determination of natural and synthetic bioactive molecules belonging to different chemical classes in complex food samples as residue of food production such as multiclass pesticides, NSAIDs or steroids, as well as a new food constituents (e.g., antioxidants) in edible plants cultivated in different countries. The use of modern spectroscopic techniques such as MS as detection system allows the identification and accurate study of the structure of all components occurring in food matrices.

While thin-layer chromatography coupled to densitometry and mass spectrometry could be the most suitable technique for preliminary screening and determination the antioxidant properties (TLC-DPPH) of food components.

Gas chromatographic methods (GC-EI-MS, GC-EI-MS/MS) are also essential for the screening of different bioactive compounds including the pesticides and fatty acids in edible plants and in food products. Pesticides profiling in food samples done by HPLC and GC in combination with prior sample separation by means of modern microextraction systems can be valuable in rapid quality control of food and ensures food use safety.

**Author Contributions:** W.P., M.D. and A.P.-P. have collected the data, designed, and written the manuscript; A.P.-P., W.P. and M.D., have revised the manuscript. All authors have read and agreed to the published version of the manuscript.

**Funding:** This research received no external funding.

**Institutional Review Board Statement:** Not applicable.

**Informed Consent Statement:** Not applicable.

**Data Availability Statement:** Not applicable.

**Conflicts of Interest:** The authors declare that there is no conflict of interests regarding the publication of this paper.

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
