# Peer review of "Current Strategies for Studying the Natural and Synthetic Bioactive Compounds in Food by Chromatographic Separation Techniques"

_processes, doi:10.3390/pr9071100_

Round 1

Reviewer 1 Report

Dear Authors,  

the article entitled »Current strategies for studying of natural and synthetic bioactive compounds by chromatographic separation techniques« summarizes the new strategies including advanced equipment and validation parameters of liquid and gas chromatography methods i.e. thin-layer chromatography, column liquid chromatography and gas chromatography suitable for identification and quantitative determination of numerous natural and synthetic bioactive.

Although the topic is very interesting and many recent studies were included in this review, my main concerns are:

  • The included compounds are many with different chemical and physical properties. Therefore, the authors should describe the different methods used by the group of the chemical compound. In this way also the manuscript will be more sorted and easier to read.
  • There are many sentences in the manuscript where the reference is missing.
  • After each chapter the sum up is missing mostly regarding the advantages and disadvantages of the methods and future prospective.
  • In my opinion the most advanced techniques should be pointed out.

 In addition, and in more detail, other concerns are included directly in the manuscript. 

Overall, mine suggestion is that the manuscript would be reconsider after major revision. Substantial changes about the results and discussion should be carried out before acceptance.

Author Response

Review 1

Dear Reviewer, we would like to thank for your consideration of this manuscript and all valuable remarks as well as advices which helped to improve the overall quality of this work.

Thank you very much also for all  important suggestions  concerning the chapter No. 3  titled Column Liquid Chromatography. We have made all these indicated corrections to this manuscript. All introduced changes/corrections are presented in yellow color. We hope that they will meet with your requirements.

Responses of the Authors

Comment 1

Dear Authors,  

the article entitled »Current strategies for studying of natural and synthetic bioactive compounds by chromatographic separation techniques« summarizes the new strategies including advanced equipment and validation parameters of liquid and gas chromatography methods i.e. thin-layer chromatography, column liquid chromatography and gas chromatography suitable for identification and quantitative determination of numerous natural and synthetic bioactive.

Although the topic is very interesting and many recent studies were included in this review, my main concerns are:

The included compounds are many with different chemical and physical properties. Therefore, the authors should describe the different methods used by the group of the chemical compound. In this way also the manuscript will be more sorted and easier to read.

Responses of the Authors

Dear Reviewer, when discussing individual chromatographic techniques, subchapters were introduced. These subsections take into account the tested groups of compounds and the matrices in which these compounds were determined.

In addition, the tables have been supplemented with information on groups of chemical compounds.

Comment 2

There are many sentences in the manuscript where the reference is missing.

Responses of the Authors

Many sentences have supplemented the references and marked them in yellow in the text.  In total, these sentences were supplemented with eighteen references.

Comment 3

After each chapter the sum up is missing mostly regarding the advantages and disadvantages of the methods and future prospective.

Responses of the Authors

Dear Reviewer, each chapter ends with an extensive summary and the advantages, disadvantages, and development possibilities of the various chromatographic techniques are discussed. The entered summaries are marked in yellow in the text.

Comment 4

In my opinion the most advanced techniques should be pointed out.

Responses of the Authors

The summaries below each chapter describe the importance of the most advanced chromatographic techniques and detection methods for the analysis of natural and synthetic bioactive compounds.

Comment 5

In addition, and in more detail, other concerns are included directly in the manuscript. 

Responses of the Authors

All comments that were included by the Reviewer directly in the manuscript were included in the revised manuscript. The introduced changes and additions are marked in yellow in the text.

We respond in detail to the following comments in the manuscript:

Comment A

Page 15: line 444: Divide the chapter by subchapters, by the each group of bioactive compound. And the description of each study should be shorter. Exclude the data that are not really important like the name of the column used.

Responses of the Authors

The suggested subchapters  in case of Column Liquid Chromatography i.e. no.  3.1, 3.2 and 3.2  were introduced in current text and similarly the data given in Tables 3-5 are  presented in accordance with chemical structure of described bioactive compounds and group division. Additional and not important data given in previous manuscript  to describe in text a proper LC procedure like for example the full name of  applied column and its size, as well as additional validation parameters of a proper LC procedure have been reduced as it was possible.

Comment B

Page 15: line 452. Reference are missing.

Responses of the Authors

Dear Reviewer, the suggested references have been introduced  in indicated place.

Comment C

Page 36, line 1081, add coupled with MS/MS or DAD etc

Responses of the Authors

Dear Reviewer, the suggested detectors have been introduced  in indicated place.

Comment D

Page 36, line 1083: Summarize what are the advantages and disadvantages of this methods, for which matrices are the most suitable. Which are the most novel methods etc.

Responses of the Authors

Extensive conclusions concerning chapter 3 titled Column Liquid Chromatography including novelties, advantages and disadvantages of described LC procedures are presented at the end of chapter 3.

We hope that it will meet with your requirements.

Sincerely yours,

Alina Pyka-PajÄ…k

Wioletta Parys

Małgorzata Dołowy

Reviewer 2 Report

The review by Parys et al. concerns with the current strategies to identify bioactive molecules from several matrices employing chromatographic methodologies.

The review is well written, but I feel that the authors misses an important branch, i.e. LC-NMR hyphenated techniques, which can provide a wealth of structural information, expecially on unknown systems.

As such, I'd like for the authors to add a paragraph on the topic. 

Author Response

Review 1

Dear Reviewer, we would like to thank for your consideration of this manuscript and all valuable remarks as well as advices which helped to improve the overall quality of this work.

We have made all these indicated corrections to this manuscript. All introduced changes/corrections are presented in yellow color. We hope that they will meet with your requirements.

Responses of the Authors

Comment

The review by Parys et al. concerns with the current strategies to identify bioactive molecules from several matrices employing chromatographic methodologies.

The review is well written, but I feel that the authors misses an important branch, i.e. LC-NMR hyphenated techniques, which can provide a wealth of structural information, expecially on unknown systems.

As such, I'd like for the authors to add a paragraph on the topic. 

Responses of the Authors

The use of combined LC-NMR techniques based on ten references is described in Chapter 6 of 'Combined Techniques'. We supplemented this chapter with three paragraphs, the text of which is marked in yellow.

We hope that it will meet with your requirements.

Sincerely yours,

Alina Pyka-PajÄ…k

Wioletta Parys

Małgorzata Dołowy

Reviewer 3 Report

The article covers the analysis of natural and synthetic chemicals present in different matrix during last three years period. It is a quite comprehensive compilation of published articles and can be useful for some readers.

However, conclusions drawn from this compilation are quite obvious and are not the discovery of last three years. It was documented and published many times that chromatographic techniques are best methods of choice for evaluation of chemicals in different matrix. This paper in just confirmation of previous findings.

In my opinion the paper is too long. Some descriptions e.g. introduction are obvious and repeating well documented knowledge. Authors should carefully consider condensation of the article and avoid repeating of information text/tables.

Lines 46-47: I completely disagree with the statement “Secondary bioactive substances are metabolic products that are not assigned basic functions in plant physiological processes”. This is an old fashion understanding of the role of natural products in plants. Also division between primary and secondary metabolism is not justified anymore.

The article can be published but after substantial reduction of its volume.

Author Response

Authors answers to Review 3

Dear Reviewer, we would like to thank for your consideration of this manuscript and all valuable remarks as well as advices which helped to improve the overall quality of this work.

Thank you very much also for all  important suggestions  concerning the conclusions, long of paper and the definition of secondary bioactive substances. We have made all these indicated corrections to this manuscript. All introduced changes/corrections are in change tracking mode. We hope that they will meet with your requirements.

Responses of the Authors

Comment 1

The article covers the analysis of natural and synthetic chemicals present in different matrix during last three years period. It is a quite comprehensive compilation of published articles and can be useful for some readers.

Responses of the Authors

We fully agree with the Reviewer's opinion that the article may be useful to some readers, especially those who deal with the analysis of pharmaceuticals and food analysis.

Comment 2

However, conclusions drawn from this compilation are quite obvious and are not the discovery of last three years. It was documented and published many times that chromatographic techniques are best methods of choice for evaluation of chemicals in different matrix. This paper in just confirmation of previous findings.

Responses of the Authors

We fully agree with the comment 2. However, the comments for each chapter were introduced as suggested by the Reviewer No. 1. These comments contain information on the advantages and disadvantages of individual methods. In our opinion, these comments and the all article will be particularly useful for young and less experienced scientists in research performed using chromatographic techniques. However, we have made some improvements to the commentaries under the individual chapters. In addition to this the final Conclusions and Abstract have been also modified.

Comment 3

In my opinion the paper is too long. Some descriptions e.g. introduction are obvious and repeating well documented knowledge. Authors should carefully consider condensation of the article and avoid repeating of information text/tables.

Responses of the Authors

Dear Reviewer, we fully agree with this suggestion and we have revised the manuscript carefully and significantly reduced its volume i.e. the introductions to the individual chapters of the article, as well as the main content of the article. Among other things, we removed those text fragments that were repeated with the contents of the tables. Since the title of the special issue is "Applications of Chromatographic Separation Techniques in Food and Chemistry" we have also removed all the chapters on biological samples. The manuscript text was 79 pages long. After reduction, thus currently it has about 50 pages. We hope that it will meet with Reviewer’s requirements.

Comment 4

Lines 46-47: I completely disagree with the statement “Secondary bioactive substances are metabolic products that are not assigned basic functions in plant physiological processes”. This is an old fashion understanding of the role of natural products in plants. Also division between primary and secondary metabolism is not justified anymore.

Responses of the Authors

Dear Reviewer, thank you for this valuable comment. As the text has been significantly reduced, the information on this subject has also been removed from the introduction, i.e. lines 43 to 60 have been removed.

We hope that it will meet with your requirements.

Sincerely yours,

Alina Pyka-PajÄ…k

Wioletta Parys

Małgorzata Dołowy

Round 2

Reviewer 1 Report

All suggestions were now taken into the account.

Overall, mine suggestion is that the manuscript would be now acceptable for publication.

Good luck!

Author Response

Authors answers to  Review 1

Dear Reviewer, thank you once again for your comments on the original version of our article. We are very pleased that you are satisfied with the revised version of our article. Thank you for your opinion that our article may be published in its current form.

We hope that it will meet with your requirements.

Sincerely yours,

Alina Pyka-PajÄ…k

Wioletta Parys

Małgorzata Dołowy
